# CD99 Expression in Glioblastoma Molecular Subtypes and Role in Migration and Invasion

**DOI:** 10.3390/ijms20051137

**Published:** 2019-03-06

**Authors:** Lais C. Cardoso, Roseli da S. Soares, Talita de S. Laurentino, Antonio M. Lerario, Suely K. N. Marie, Sueli Mieko Oba-Shinjo

**Affiliations:** 1Laboratory of Molecular and Cellular Biology (LIM 15), Department of Neurology, Faculdade de Medicina FMUSP, Universidade de Sao Paulo, Sao Paulo 01246-903, Brazil; laisccardoso@usp.br (L.C.C.); roselis@usp.br (R.d.S.S.); talitalaurentino@usp.br (T.d.S.L.); sknmarie@usp.br (S.K.N.M.); 2Department of Internal Medicine, Division of Metabolism, Endocrinology, and Diabetes, University of Michigan, Ann Arbor, MI 48109, USA; alerario@umich.edu

**Keywords:** glioblastoma, CD99, migration, cytoskeleton, transcriptome

## Abstract

Glioblastoma (GBM) is the most aggressive type of brain tumor, with an overall survival of 17 months under the current standard of care therapy. CD99, an over-expressed transmembrane protein in several malignancies, has been considered a potential target for immunotherapy. To further understand this potentiality, we analyzed the differential expression of its two isoforms in human astrocytoma specimens, and the CD99 involved signaling pathways in glioma model U87MG cell line. CD99 was also analyzed in GBM molecular subtypes. Whole transcriptomes by RNA-Seq of *CD99*-siRNA, and functional in vitro assays in *CD99*-shRNA, that are found in U87MG cells, were performed. Astrocytoma of different malignant grades and U87MG cells only expressed CD99 isoform 1, which was higher in mesenchymal and classical than in proneural GBM subtypes. Genes related to actin dynamics, predominantly to focal adhesion, and lamellipodia/filopodia formation were down-regulated in the transcriptome analysis, when *CD99* was silenced. A decrease in tumor cell migration/invasion, and dysfunction of focal adhesion, were observed in functional assays. In addition, a striking morphological change was detected in *CD99*-silenced U87MG cells, further corroborating CD99 involvement in actin cytoskeleton rearrangement. Inhibiting the overexpressed CD99 may improve resectability and decrease the recurrence rate of GBM by decreasing tumor cells migration and invasion.

## 1. Introduction

Glioblastoma (GBM) is the most common malignant brain tumor [1]. The 2016 World Health Organization (WHO) added molecular parameters to histological characteristics to stratify brain tumors [2]; however, only a few molecular targets have shown potential as predictive factors for prognosis or response to therapy [3]. The standard of care with radiotherapy and temozolomide for GBM patients has led to an overall survival period of only 15 to 17 months [4,5]. Such dismal results are due to the highly invasive nature of GBM cells in the normal brain parenchyma, preventing its complete surgical resection and inexorable tumor recurrence [6]. Therefore, new therapeutic approaches for GBM patients are required. The inhibition of constitutively activated signal transduction pathways has demonstrated considerable potential to better control tumor growth [7]. Additionally, immunotherapy has been demonstrated as another approach [8], and CD99, a tumor-associated antigen, is a candidate target for this modality of therapy [9,10].

In normal cells, the activation of CD99 leads to cell adhesion and migration, homotypic aggregation, apoptosis, expression upregulation, and the transport of various membrane proteins. Moreover, in inflammatory cells, CD99 has been related to Th1 cell differentiation and the activation and proliferation of mature T cells [11,12]. However, CD99 has been implicated as playing dual roles in tumors, as an oncogene (glioma, Ewing’s sarcoma, lymphoma/leukemia, melanoma, and breast cancer) or an oncosuppressor (osteosarcoma, Hodgkin’s lymphoma, and cancers of the stomach, pancreas, and bladder), as recently reviewed [13]. Additionally, previous microarray expression analyses in gliomas have revealed CD99 among 31 genes coding for membrane proteins potentially targetable for immunotherapy [14]. Nonetheless, the molecular mechanisms involving CD99 are still not fully understood.

*CD99* encodes two distinct proteins by alternative splicing [15]. CD99 isoform 1 comprises an extracellular domain glycosylated with O-linked sugar residues, a transmembrane domain and an intracytoplasmic domain with 36 amino acids. CD99 isoform 2 is truncated at the intracytoplasmic domain, presenting only 28 amino acids [12]. CD99 isoforms play distinct functional roles. On B lymphocytes, isoform 1 promotes cell–cell adhesion, while isoform 2 inhibits homotypic adhesion. Both isoforms are required to induce apoptosis in thymocytes and immature T cells [15]. In tumors, such as osteosarcoma, isoform 1 has been described as a potent suppressor of cell migration and invasion, in contrast to isoform 2, which plays an important role in tumor cell migration and metastatic capacity [16]. Similarly, the CD99 isoform 2 shows the enhanced invasive ability of human breast cancer cells [17].

In the present study, we examined the molecular mechanisms related to CD99 in astrocytomas, especially in GBM, based on human tumor samples and an in vitro cellular model.

## 2. Results

### 2.1. CD99 Isoforms Expression in Human Astrocytomas and in U87MG Cell Line

Striking predominant expression of isoform 1 was observed in different grades of astrocytoma (I-IV) (Figure 1a), and the expression level was higher in astrocytoma samples, compared to that in non-neoplastic (NN) brain tissue, with higher expression in GBM samples. No difference was found in pairwise comparisons of different grades of astrocytoma. The *CD99* expression evaluated in 37 classical, 14 mesenchymal, and 14 pro-neural GBM samples from the present cohort [18] showed lower, although not significant, expression in the proneural subtype (Figure 1b). In a larger GBM cohort from The Cancer Genome Atlas (TCGA) database, with 38 classical, 53 mesenchymal, and 29 proneural subtype samples, a significantly higher expression of *CD99* in classical and mesenchymal subtypes, than that in proneural subtypes, was observed (Figure 1c). Additionally, the expression analysis of *CD99* isoforms, in the U87MG cell line, confirmed the presence of only isoform 1 (Figure 1d), which was also confirmed at the protein level by western blotting with the detection of a unique band of 32 kDa (Figure 1e).

### 2.2. Transcriptome Analysis of CD99-siRNA U87MG

The differential expression analysis of the U87MG knockdown for *CD99* and negative non-target control NTC (CD99-siRNA vs. NTC-siRNA) resulted in 2,828 genes, presenting statistical expression differences with adjusted *p* ≤ 0.01. CD99 presented the highest fold-change (4.19, corresponding to a 17.51-fold decrease), confirming the efficiency of CD99 gene silencing. The enrichment analysis by DAVID algorithm showed two enriched clusters of functional annotation (Figure 2a), with the first cluster related to cell adhesion. Then, we further investigated the specific genes associated with this cluster encoding membrane, extracellular matrix, stress fiber, focal adhesion, and filopodia/lamellipodia proteins. A heatmap, and the differences in gene expression (fold-change) of these selected genes, are represented in Figure 2b,c, respectively. *OPN* (osteopontin), and *LAMA5* (laminin 5) were downregulated, while *FN1* (fibronectin 1), *THBS1* (thrombospondin 1), and *COL6A2* (alpha(α)2(VI) chain of type VI collagen) were upregulated when *CD99* was silenced. The genes encoding transmembrane proteins, which interact with extracellular matrix proteins, such as *CD44* and some integrin subunits (*ITGB8*, *ITGA2*, *ITGAV*, and *ITGA5*), were also upregulated.

### 2.3. Functional Analysis of CD99 Involvement in Glioma Cell Migration, Invasion and Adhesion

*CD99*-siRNA silencing was transitory, and after 7 days, its gene expression was recovered to approximately 50% of control NTC (data not shown). Permanent long-term CD99 knockouts were constructed to further study the GBM cell phenotype. The two *CD99*-shRNA constructs significantly reduced CD99 expression to 20 and 3% when compared to the control cells (scrambled) (Figure 3a). These results were validated at the protein level by western blotting (Figure 3b).

The U87MG cell migration after CD99 knockout showed a significant decreased migratory activity compared to that of the controls (*p* < 0.05) (Figure 3c–e) by two image recording methods in a period of 24 h. Particularly, shCD99-2 caused a stronger effect in migration than shCD99-1 (Figure 3c,e). The difference between the two shRNA constructs was evident after 6 h in the monitored migration assay.

An inhibitory invasion effect of 47% for shCD99-1 and 45% for shCD99-2 in relation to scramble was observed (*p* < 0.05 for both analysis) (Figure 4a). The short-term adhesion assay for 3 h demonstrated the reduced adhesion of shCD99-1, compared to control, in contrast with the increased adhesion of shCD99-2 in relation to control (*p* < 0.0001 for both comparisons) (Figure 4b). A residual 20% of CD99 expression in shCD99-1 may explain the opposing findings in this assay.

### 2.4. CD99 Colocalizes with F-Actin

The colocalization of CD99 with phalloidin was observed in the cell-cell contact regions and in lamellipodia in control U87MG cells, and CD99 expression was not detected in U87MG cells knocked down for CD99, as expected. Interestingly, relevant changes in morphology were observed in both U87MG shCD99-1 and 2 cells, consisting of the altered organization of actin microfilaments and cytoplasmic protrusions compared to the controls. Moreover, actin distribution was more homogeneous in all adherent membrane cells in shCD99-1, while more spaced actin bundles were shown in shCD99-2 (Figure 4c).

## 3. Discussion

CD99 has been described as a highly expressed tumor-associated antigen in GBM tissues and a potential target for synthetic multi-peptide vaccines or dendritic cell immunotherapy [10]. Nonetheless, the functional role of CD99 in tumors has not yet been fully elucidated, and the ligand for CD99 has not yet been identified. Thus far, the current knowledge of CD99 function derives from experiments of CD99 activation by agonist monoclonal antibodies, in hematopoietic and tumor cells [19,20]. In addition, the knowledge of CD99-regulated pathways in the GBM is relatively sparse.

The findings of the present study show that CD99 is a worthwhile target to be explored for therapeutic purposes.

### 3.1. Expression of CD99 Isoforms

*CD99* isoform 1 was the predominant isoform present in different malignant grades of human astrocytomas, when compared to NN brain tissue, as observed in other studies [10]. Its expression was particularly higher in GBM, when compared to lower malignant grade diffusely infiltrative astroctyomas [9]. CD99 dual behavior has been reported to be associated with tumor progression in Ewing sarcoma and acute lymphoblastic leukemia [21], as tumor suppressor in osteosarcoma [22], and in Hodgkin’s lymphoma [23], which may be attributed to differential preponderance of CD99 isoforms. In fact, CD99 isoform 1 has been reported to regulate lymphocyte adhesion mediated by the integrin LFA-1, in contrast to isoform 2, which is associated with the inhibition of spontaneous adhesion of these cells via the LFA-I/ICAM pathway [24]. In tumors, CD99 isoforms have also been related to different functions in tumor malignancy. In osteosarcoma, isoform 1 acted as a potent suppressant of migration and metastasis through c-Src repression [15,16], while isoform 2-transfected breast cancer cell lines and showed a superior ability to migrate compared to the control, and this difference was not observed in the same cells transfected with the isoform 1 [17,25]. Additionally, and antibody against CD99 induced CD99 engagement and a consequent non apoptotic, caspase-independent programmed cell death of Ewing sarcoma cells, with micropinocytosis by a pathway that resembles methuosis [26]. More recently, a ligand to CD99 extracellular region, clorafabine, inhibited malignant properties of Ewing sarcoma cells, opening new perspectives to target this molecule for cancer therapy [27]. In our GBM model, CD99 isoform 1 proved to be pro-tumorigenic, and therefore the isoform as druggable target.

### 3.2. Transcriptome Analysis and Signaling Pathways Modulated by CD99

Interestingly, the U87MG *CD99*-siRNA transcriptome analysis pointed out that the most enriched processes, that are related to the downregulation of *CD99,* were cell-cell adherent junction and adhesion in accordance with previously reported *CD99* roles in tumors. We hypothesize that CD99 may modulate integrin inside-out signaling pathway through the activation of G-protein β/γ subunit/c-Scr/FAK1/Talin. Talin binds to integrin β3 cytoplasmatic tail and induces conformational changes in their extracellular domains, resulting in an increased integrin affinity for ligands [28]. Alternatively, CD99 may activate caveolin-1 via integrin αV or α5, and caveolin-1 thus activating FAK1. Once FAK1 is activated, vinculin activates α-actin promoting cell adhesion and stress fiber formation. Moreover, FAK1 may activate CDC42/RAC that regulates Arp2/3-mediated actin polymerization and consequent orientation of cell migration [29]. Recently, another pathway has been described in a breast cancer model, where CD99-derived agonist ligands inhibit fibronectin-mediated β1 integrin activation, through the SHP2/ERK/PTPN12/FAK1 signaling pathway [30]. The suppression of β1 integrin activity by CD99 activation may occur through the dephosphorylation of FAK1 at Y397 [31].

Therefore, the present transcriptome data and the findings of previous studies suggest that extracellular matrix targets, together with tumor-associated antigens, including CD99 and integrins, participate in a cytosolic downstream pathway intermediated by FAK1 and c-Src. Actually, the analysis of focal adhesion and regulation of actin cytoskeleton pathways in the present transcriptome data demonstrated the downregulation of RhoGap expression genes (*ARHGAP5* and *ARHGAP35*), *DIAPH1,* myosin light chain genes (*MYL5*, *MYL6B* and *MYL12B*), related to the stress of fiber formation. Similarly, the downregulation of genes, which code for several targets downstream of FAK1 related to filopodia and lamellipodia formation by actin branch regulation essential for cell migration, such as Rac family small GTPase 2 (*RAC2*), cytoplasmic FMR1 interacting protein 2 (*CYFIP2*), WAS protein family member 1 (*WASF1)*, BAI1-associated protein 2 (*BAIAP2*), and actin-related protein 2/3 complex subunit 1A and 1B (*ARPC1A* and *ARPC1B*) was observed.

Additionally, genes coding ezrin-radixin-moesin (ERM) complex, named *EZR*, *RDX,* and *MSN*, respectively, were downregulated when CD99 was silenced. When the ERM complex is activated, the N-terminal, named FERM domain, binds the actin cytoskeleton to the cell membrane [32]. Therefore, a downregulation of this complex can further disorganize the actin cytoskeleton and consequently prevent cell migration. Similarly, genes coding for dynamin 2 (*DNM2*) and cortactin (*CTTN*) were also downregulated with the reduction of CD99 expression. These proteins participate in cell migration by stabilizing F-actin bundles in filopodia [33,34].

### 3.3. Influence of CD99 on Migration, Invasion and Cell Adhesion

The functional assays, with U87MG *CD99*-shRNA, demonstrated that *CD99* downregulation resulted in decreased migration and invasion of tumor cells. GBM cells use a mesenchymal mode of migration and invasion similar to fibroblasts [35,36]. Nonetheless, glioma U87MG cells over-expressing CD99 have presented a higher proportion of amoeboid migration, with high cortical tension and low adhesion to ECM, than that in control cells [37,38].

Decreased cell motility, after CD99 silencing in GBM cells, may be related to dysregulation of stress fiber formation, focal adhesion and particularly to the dysregulation of filopodia and lamellipodia formation, according to the downregulated set of genes identified by the transcriptome analysis (Figure 5). In fact, the overexpression of isoform 1 in osteosarcoma cell lines likewise modulated the expression of genes essential for remodeling the actin cytoskeleton and cell invasion through expression of *ARP2* and *ARPC1A*, which code for proteins of the Arp2/3 complex [16,39]. These previous reports corroborate the present results of the transcriptome analysis. The dysfunction of the Arp2/3 complex is an important node for the regulation of cell motility.

The functional assays, with U87MG *CD99*-shRNA, also showed the dysfunction of cell adhesion, suggesting that this function in U87MG cells is dependent on the amount of CD99 expressed in the cells. The complete or near complete knockout of CD99 (98% knockdown with shCD99-2) presented a higher number of adhered cells in relation to control, in contrast to the prevention of cell adhesion with residual CD99 expression, as observed with shCD99-1 knocked down cells (20% of residual CD99 expression). The difference may be explained by the proportion of upregulated expression of extracellular matrix components as fibronectin, when CD99 was silenced. In fact, the high concentrations of fibronectin has been related to more abundant, but less dynamic adhesions, whereas low concentrations of fibronection, resulted in few but high dynamic adhesions, and more organized pattern of actin polymerization [40]. The present transcriptome analysis showed highly upregulated expression of fibronection with CD99 silencing, which may explain the loss of cell adherence. Indeed, our immunofluorescence images, co-localized CD99 and phalloidin at the cell-cell contact regions (lamellipodia) of the plasma membrane in control U87MG cells, which were strikingly lost in the absence of CD99, leading to the retraction of the cell surface. The small amount of residual CD99, as occurred with shCD99-2 cells, preserved a more spread cell morphology compared to shCD99-1 cells (Figure 4), which may preserve cell adherence through small F-actin bundles in focal adhesion [41,42]. Previous results in Ewing’s sarcoma cells, with a CD99 agonist antibody, demonstrated the ability of CD99 to control cell cytoskeleton remodeling and formed adherent junctions and focal adhesions [43], further corroborating the present findings.

### 3.4. CD99 Is Upregulated in Glioma Cell Line U87MG

Interestingly, the TCGA GBM RNA-Seq dataset analysis showed that the most aggressive molecular subtype of GBM, the mesenchymal one, presented the highest expression of *CD99*. Functional analyses using U87MG, a GBM mesenchymal subtype model, demonstrated that reducing the expression of CD99 decreases migration and invasion of the tumor cells. Therefore, therapeutic strategies to downregulate CD99 may improve tumor respectability and may reduce the probability of tumor recurrence. Further knowledge on the downstream molecular mechanisms, that lead CD99 to modify the cellular actin dynamics, may also enable the identification of other druggable targets for combinatorial therapy and to better improve the outcome of this dreadful tumor.

## 4. Materials and Methods

### 4.1. Cell Cultures

The glioma cell line, U87MG, and the human embryonic kidney cell line, HEK373T, from the American Type Culture Collection (ATCC) were maintained in Dulbecco’s Modified Eagle’s medium (DMEM) (Thermo Fisher Scientific, Whatham, MA, USA), supplemented with 10% heat-inactivated fetal bovine serum (FBS) (Thermo Fisher Scientific), antibiotics (100 units/mL penicillin, 100 µg/mL streptomycin) in a humidified atmosphere of 5% CO_2_ in air at 37 °C. The cell lines were authenticated by short tandem repeat DNA analysis with the commercial GenePrint 10 System (Promega, Fitchburg, WI, USA).

### 4.2. Casuistry

Astrocytoma samples of different grades and non-tumoral brain tissues from epilepsy surgeries were collected during the surgical procedure and immediately frozen in liquid nitrogen after resection by a group at the Division of Neurosurgical of the Department of Neurology, School of Medicine, University of Sao Paulo (HC-FMUSP). A total of 19 NN samples and 150 tumor samples, comprised of 23 AGI, 26 AGII, 17 AGIII and 84 GBMs, were analyzed. The project was approved by the National Ethics Commission (CONEP) and the local institutional ethics committee (numbers 197/2002 and 830/2001, respectively). Post-informed consents were obtained from all patients included in the present study.

### 4.3. Total RNA Extraction and cDNA Synthesis

More than 80% of tumor tissue was ensured in all tumor samples. The RNeasy Mini Kit (Qiagen, Valencia, CA, USA) was used for total RNA extractions. RNase-free DNase (Qiagen) treatment and quality and concentration assessments were performed. cDNA was obtained by reverse transcription, using SuperScript III reverse transcriptase, RNase inhibitor (RNaseOUT), random oligonucleotides and oligo dT, according to the manufacturer’s recommendations (Thermo Fisher Scientific). After treatment with 1 U RNase H (Thermo Fisher Scientific) at 37 °C for 30 min and at 72 °C for 10 min, cDNA was diluted in Tris-EDTA buffer and stored at −20 °C for subsequent analysis.

### 4.4. Quantitative Real Time PCR

*CD99* expression levels in tissue samples and cell lines were analyzed by quantitative real time PCR (qRT-PCR) by the SYBR Green method, on ABI 7500 apparatus (Thermo Fisher Scientific). Table 1 shows the sequences of primers for CD99 isoforms 1 and 2 and reference genes synthesized by IDT (Coralville, IA, USA). All the reactions were performed in triplicate. The final reaction volume of each reaction was 10 μL, and contained 2.5 μL of cDNA, 5 μL of Power SYBR Green PCR Master Mix (Thermo Fisher Scientific) and 2.5 μL of primers in a pre-standardized concentration. The amplification conditions included an initial incubation at 50 °C for 5 min and 95°C for 10 min, followed by 40 cycles at 95 °C for 15 s and 60 °C for 60 s. The expression value of *CD99* was normalized with reference genes as the internal controls: *HPRT*, for cell experiments, and *HPRT*, *GUSB*, and *BCRP*, for tissue analyses. For cell analysis, *HPRT* was used as the reference gene. Single product amplification was confirmed by analyzing its dissociation curve. The amplification efficiencies [E=10^(-1/slope)^-1)] were calculated using serial cDNA dilutions. Equation 2^−∆*C*t^ was applied in the calculation of the relative gene expression for efficiency (E) = 100 ± 10%, where ∆*C*t = [mean *C*t of CD99)]—[mean *C*t of HPRT or geometric mean of mean Ct of housekeeping genes] [44].

### 4.5. CD99-siRNA and Library Preparations for NGS Sequencing (RNA-Seq)

*CD99* siRNA was performed as previously described in experimental duplicates [9]. The libraries were constructed with the TruSeq Stranded Total RNA kit (Illumina, San Diego, CA, USA) and quantified by qRT-PCR using Kapa Library Quantification Kit (Kapa Biosystems, Wilmington, MA, USA). The mean size of each library was determined on the Tapestation 2200 (Agilent Technologies, San Jose, CA, USA). RNA-Seq sequencing was performed on a HiSeq 2500 (Illumina) sequencer in the SELA Facility Core of School of Medicine, University of Sao Paulo.

### 4.6. Transcriptome Analysis

Sequencing generated an average of 51 million reads per samples. Quality control analysis was performed by FASTQC software. Raw reads were aligned to the hg38 through STAR software [45]. Quantification of the gene expression data was performed through Feature Counts software [46,47], and the data were normalized according to two different methods: Reads per kilobase million (RPKM) and counts per million (CPM) [48]. Differential expression analysis (bioconductor portal) was analyzed by the Limma-voom framework [49]. The raw data were initially log-transformed and normalized. Subsequently, differential expression between groups was analyzed by linear models and the application of moderate t statistics. This analysis was performed with the RNA-Seq tool [50]. Finally, the analysis of RNA-Seq data (genes differentially expressed in cells silenced by siRNA compared with the NTC) was performed through the DAVID database [51,52] functional and pathway enrichment. RPKM values were transformed to z-scores for heatmap visualization [53].

### 4.7. TCGA Data Analysis

The TCGA GBM gene expression from RNA-Seq dataset were downloaded (Genomics Data Commons Data Portal) [54] and normalized by DEseq R software.

### 4.8. Construction of Recombinant Lentivirus

Two *CD99* shRNA molecule (MISSION shRNA Library, Sigma-Aldrich, St Louis, MO, USA) sequences were as follows (5′–3′): CCGGGCGCGATAGCGCTAATAATTTCTCGAGAAATTATTAGCGCTATCGCGCTTTTT-3′, for control (scrambled), CCGGGCGTTTCAGGTGGAGAAGGAACTCGAGTTCCTTCTCCACCTGAAACGCTTTTTG (shCD99-1), and CCGGCGGATGGTGGTTTCGATTTATCTCGAGATAAATCGAAACCACCATCCGTTTTTG (shCD99-2). The lentivirus transduction particles were prepared by the co-transfection of 2 × 10^6^ HEK293T cells, with 9 µg of pCMV-deltaR8.2 containing Gag and Pol genes, 0.9 µg of pCMV-VSG-G expressing the G-protein of the vesicular stomatitis, and 9 µg of pLKO.1 bearing the shRNAs. For transfection, the reagent FuGENE HD (Promega) was added in antibiotic-free DMEM with 10% FBS. Lentiviral particles in the culture medium were collected at 48 h and 72 h, centrifuged, filtered (0.4 µm), aliquoted and stored at −80 °C until further use.

### 4.9. Generation of Knockdown Cell Lines

A total of 1 × 10^5^ U87MG cells were infected with scrambled 1, scramble 2, shCD99-1 and shCD99-2 with DMEM containing 8 µg/mL of polybrene. At 24 h after transduction, the supernatant was removed and replaced with culture medium containing puromycin. The *CD99* knockdown efficacy was assessed by qRT-PCR and Western blot analyses.

### 4.10. Western Blotting

Cell protein extracts were obtained with urea/Chaps lysis buffer and protease inhibitor cocktail (Sigma-Aldrich). Total protein concentrations were determined by the Pierce BCA Protein Assay Kit (Thermo Fisher Scientific). Cell lysates (30 μg of proteins) were separated by 4–12% gradient polyacrylamide gel electrophoresis (Thermo Fisher Scientific) in NuPAGE MOPS SDS electrophoresis buffer (Thermo Fisher Scientific) and transferred to a nitrocellulose membrane through the iBLOT system (Thermo Fisher Scientific). The membrane was incubated with mouse monoclonal anti-CD99 (1:1000, DN16, Abcam, Cambridge, MA, USA) and mouse monoclonal anti-β-actin (1:20,000, clone AC-74, Sigma-Aldrich) as control for protein loading. A secondary antibody anti-mouse IgG (1:1000, Sigma-Aldrich) conjugated to peroxidase and the chemiluminescence detection system (Western Lightning Plus-ECL, Enhanced Chemiluminescence Substrate, Perkin Elmer, Whaltham, MA, USA) were used to visualize proteins in the membrane on the ImageQuant LAS4000 apparatus (GE Healthcare, Pittsburgh, PA, USA).

### 4.11. Wound Healing Migration Assay

A total of 2 × 10^5^ cells/well of silenced cells and their respective controls (scrambled) were grown on 24-well plates, previously coated with Poly-l-Lysine (Sigma-Aldrich), until the cells reached confluence. The culture medium was removed, and a discontinuity cell-free area was formed by scrapping the monolayer with a micropipette tip. Debris were removed by washing with PBS and then replacing it with 2 mL of fresh medium supplemented with 1% FBS. Images from three points along each wound were selected and acquired at different time intervals (zero, 6, 12, and 24 h). The assays were performed in quadruplicates, and in two independent assays. In parallel, a real-time assay was performed on Axiovert 200M Inverted Motorized Microscope (Carl Zeiss, Jena, Germany) and wound closure was recorded every 3 h for 24 h. The assay was performed in duplicate, and 3 different fields of each well were photographed. For both experiments, the cell-free areas were initially calculated as a percentage of the initial cell free area (time zero) and arbitrarily labeled 100%. The percentage of the cell free area was calculated by ImageJ [55]. The percentage of migrated area at different times was determined by the difference of the area at time zero (100%) minus the area free of cells [55].

### 4.12. Invasion Assay

Invasion chambers were used according to the manufacturer’s instructions (BD BioCoat Matrigel invasion assay; BD Biosciences, San Jose, CA, USA). U87MG cells (shCD99-1 and 2 and controls) were maintained for 2 h in DMEM supplemented with 1% FBS. A total of 2.5 × 10^4^ cells was suspended in 0.5 mL DMEM, supplemented with 1% FBS and seeded onto the upper compartment of the Matrigel-coated trans-well inserts fitting the 24-well plates. The bottom chamber was filled with 10% FBS as a chemo-attractant. Cells were incubated at 37 °C for 18 h. Non-invading cells were wiped away from the upper surface of the chamber. Invading cells were fixed with 4% paraformaldehyde, stained with crystal violet (0.2% in methanol 20%) and analyzed by inverted microscopy with 10× magnification. The results were quantified by counting all the cells of the inserts in duplicate from two independent experiments. The invasion values were expressed as the percentage of invaded cells in relation to the control.

### 4.13. Adhesion Assay

U87MG shCD99-1, shCD99-2, and controls were incubated for 2 h in DMEM supplemented with 1% FBS. A total of 5 × 10^4^ cells were allowed to adhere to 96-well plates for 3 h at 37 °C in 5% CO_2_. The cells were gently washed three times with PBS, and attached cells were quantified after incubation with PrestoBlue Cell Viability Reagent (Thermo Fisher Scientific) for 2 h at 37 °C in 5% CO_2_ in a humidified atmosphere, by measuring the resulting fluorescent signal at 525 nm by using the GloMax—Multi Microplate Multimode Reader (Promega).

### 4.14. Immunofluorescence

For this analysis, 2 × 10^4^ U87MG cells were plated onto the coverslips, previously coated with Poly-L-Lysine (Sigma-Aldrich) and allowed to attach. After 24 h, the cells were fixed in 4% paraformaldehyde in PBS for 90 min at 4°C. Subsequently, the cells were washed 3 times with PBS for 5 min and permeabilized with 0.1% NP-40 (Abcam) in PBS for 30 min at 37 °C, followed by two washes with PBS for 5 min. Non-specific sites were blocked by incubating cells with 4% goat serum (Sigma-Aldrich) in PBS (block solution) for 30 min at 37 °C. The cells were incubated with the primary antibody, anti-CD99, conjugated with FITC (1:50, Thermo Fisher Scientific) and phalloidin labeled with Alexa Fluor 488 (1:50, Thermo Fisher Scientific) diluted in block solution. After incubation for 48 h at 4 °C, and three washes with PBS for 5 min, the nuclei were counterstained with DAPI (1:1000, Thermo Fisher Scientific) by incubating for 3 min. Finally, the samples were washed 3 times with PBS for 5 min and embedded with mounting medium ProLong Gold Antifade Mountant (Thermo Fisher Scientific). Negative controls include complete reaction and absence of primary antibody. The documentation and analysis of the slides were performed with the LSM 510 Meta Confocal System (Carl Zeiss).

### 4.15. Statistical Analysis

The normality test of Kolmogorov-Smirnov and Shapiro-Wilk test were applied to analyze the distribution of the *CD99* gene expression data by qRT-PCR. Non-parametric Kruskal-Wallis and Dunn’s tests were used to analyze the differences in *CD99* expression among all groups, between NN and each group of astrocytoma of different grades or between NN and GBM molecular subtypes. The cell data were expressed as the average of the values ± standard deviation. The Student’s t test was used to calculate the statistical significance of *CD99* knockdown expression in U87MG cell line in relation to scramble. The wound healing results of cells that were silenced for *CD99* and controls were performed by Two-way Anova, followed by post-hoc Bonferroni test. A non-linear regression test was performed for the real-time migration assay to compare groups. For invasion and adhesion assays, comparisons between cells transduced with shCD99-1 and shCD99-2 and respective controls were performed by using Mann-Whitney test. For real-time migration assays, non-linear regression analysis was used. Differences were considered statistically significant when *p* < 0.05. The calculations were performed using GraphPad Prism v.7.0 (San Diego, CA, USA) and SPSS software, v.15.0 (Chicago, IL, USA).

## 5. Conclusions

In conclusion, the present study showed that isoform 1, of CD99, is exclusively present in human astrocytomas and the U87MG cell line and regulates functions, such as cytoskeleton remodeling, cell migration, invasion, and adhesion. The present transcriptome data of *CD99* silenced GBM cells, suggest that CD99 modulates FAK1/c-Src signaling pathways, related to actin cytoskeleton dynamics. The downregulation of this pathway through CD99 knockout, may enable the regulation of the migration and invasion of GBM cells, particularly in the mesenchymal subtype, and may increase the clinical outcome, thereby improving tumor resectability and decreasing the tumor recurrence rate.

## Figures and Tables

**Figure 1 ijms-20-01137-f001:**
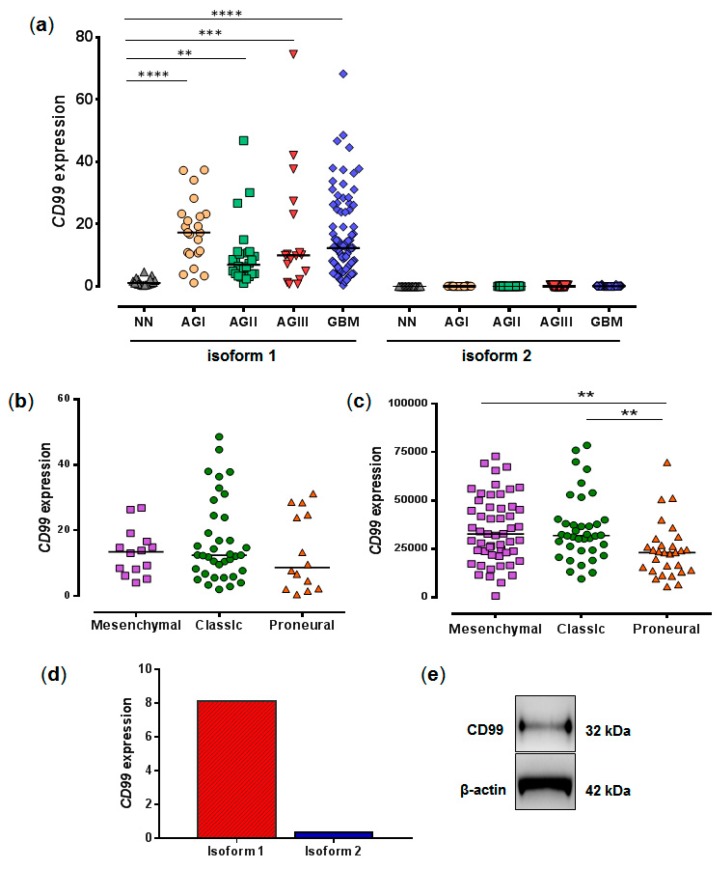
Expression of *CD99* in astrocytomas of different malignant grades and the U87MG cell line. (**a**) Relative quantification of mRNA of *CD99* isoforms 1 and 2 in 23 pilocytic astrocytoma (AGI), 26 low-grade astrocytoma (AGII), 17 anaplastic astrocytoma (AGIII), 84 glioblastoma (GBM), and 19 non-neoplastic (NN) tissue samples. The relative expression values were calculated, based on the geometric mean of the *HPRT*, *BCRP*, and *GUSB* housekeeping expression levels of each sample. The differences among the groups were significant (*p* < 0.0001, Kruskal-Wallis test). The horizontal bar indicates the median of each group. Asterisks indicate statistical differences: ** *p* < 0.01, *** *p* < 0.001, **** *p* < 0.0001, Dunn’s test. (**b**) Isoform 1 CD99 expression levels of GBM molecular subtypes in the present series determined by qRT-PCR and (**c**) in TCGA database determined by RNA-Seq. Differences among groups were significant (*p* = 0.0031, Kruskal-Wallis test) for TCGA cases (** *p* < 0.01 for proneural vs. classic and proneural vs. mesenchymal, Dunn’s test). The horizontal bar indicates the median of each group. (**d**) Relative quantification of mRNA for CD99 isoforms 1 and 2 in glioma cell line U87MG. *HPRT* was used as a reference gene. The results were expressed as the means of 2 independent experiments. (**e**) Representative western blot, showing the expression of CD99 in U87MG. β-actin was used as a control in the experiment. Only one band, corresponding to isoform 1 with 32 kDa, was observed.

**Figure 2 ijms-20-01137-f002:**
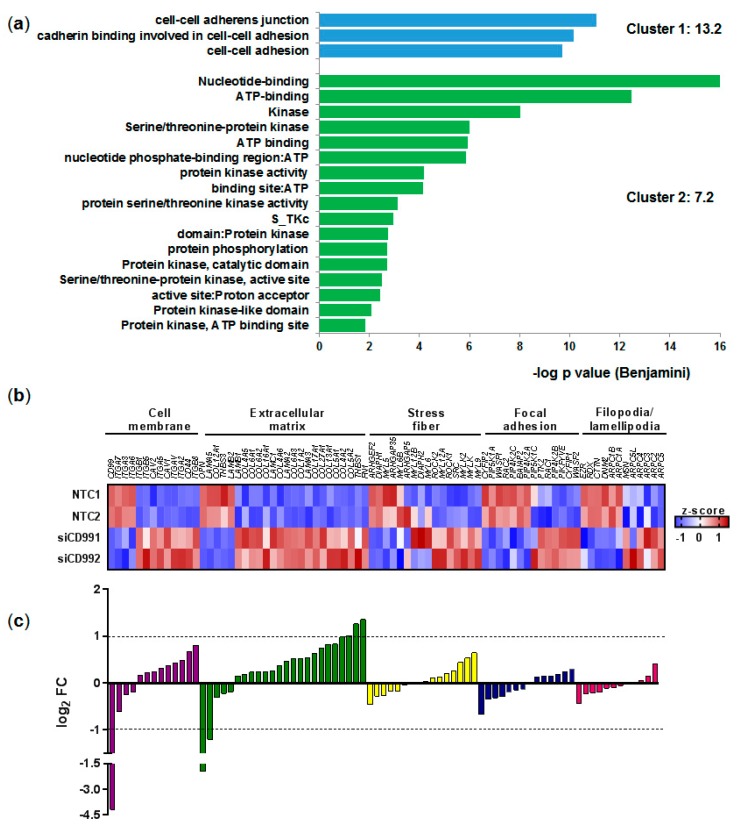
Transcriptome analysis of U87MG cell line knockdown for *CD99* by siRNA. (**a**) The two most enriched functional annotation clusters by the DAVID annotation cluster analysis of differentially expressed genes after *CD99* knock down (adjusted *p* ≤ 0.01). (**b**) Heatmap representing expression of genes coding for proteins of cell membrane, extracellular matrix, stress fiber, focal adhesion and filopodia/lamellipodia. RPKM values of experimental duplicates were normalized by z-score (**c**) The log_2_ fold-change (FC) of genes represented in the heatmap.

**Figure 3 ijms-20-01137-f003:**
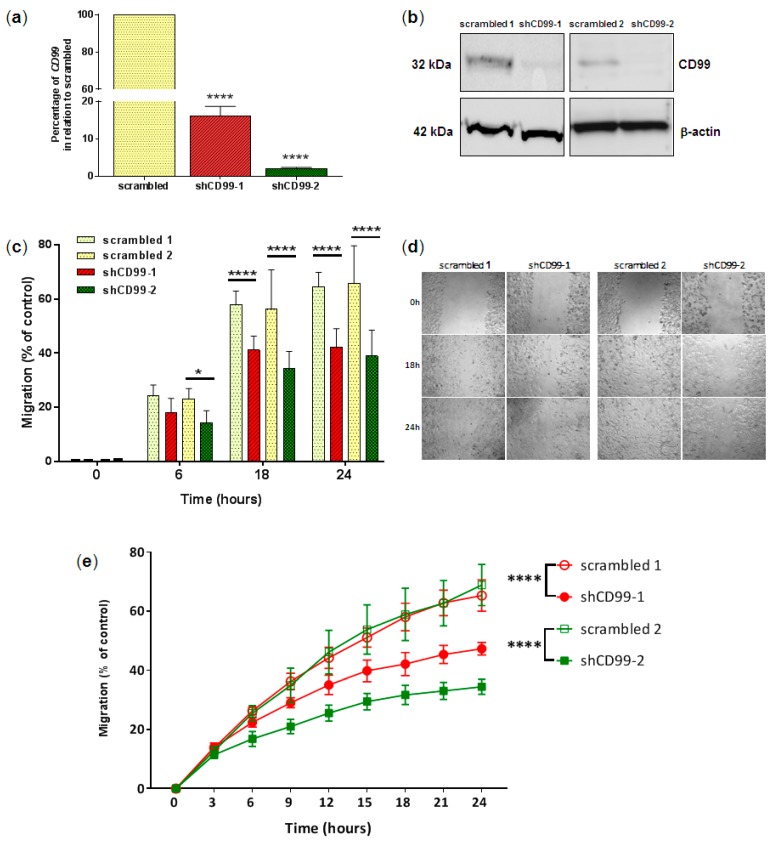
Effects of *CD99* knockdown by shRNA on U87MG cell line on migration. (**a**) Quantification of the mRNA levels in U87MG cells for CD99 knockdown (shRNA) and control (scrambled) for two clones (shCD99-1 and shCD99-2). *HPRT* was used as a reference gene for the analysis. Graphs represent 3 independent experiments. The asterisks indicate statistically significant differences of shCD99-1 and shCD99-2 and their respective controls (**** *p* < 0.0001, Student’s *t* test). (**b**) Western blot analysis of the CD99 expression in U87MG cell line after transduction with shCD99-1 and shCD99-2 and controls (scrambled). β-actin was used as control for protein loading. (**c**) Migration assay, showing that U87MG cells transfected with shCD99-1 and shCD99-2 with reduced migratory activity compared to that of controls. The graph represents the area invaded by cells at 0, 6, 18 and 24 h after scratching. The results were expressed as the means ± standard deviation (quadruplicates) of two independent experiments. Asterisks indicate statistically significant differences of the CD99 silenced cells in relation to their controls at different times (* *p* < 0.05 and **** *p* < 0.0001, Bonferroni test). (**d**) Representative photomicrographs of wound closure of U87MG cell line at 0, 6, 18, and 24 h, 10× magnification. (**e**) Real-time migration assay, confirming reduced migration observed each 3 h for a period of 24 h. U87MG cells knocked down with shCD99-1 (closed circle) and shCD99-2 (closed square) showed reduced migratory activity, when compared to respective controls, scrambled 1 (open circle) and scrambled 2 (open square). The result was expressed as the means ± standard deviation (sextuplicate). The asterisk indicates statistically significant difference of cells silenced for *CD99* in relation to scrambled at different times (**** *p* < 0.0001; non-linear regression analysis).

**Figure 4 ijms-20-01137-f004:**
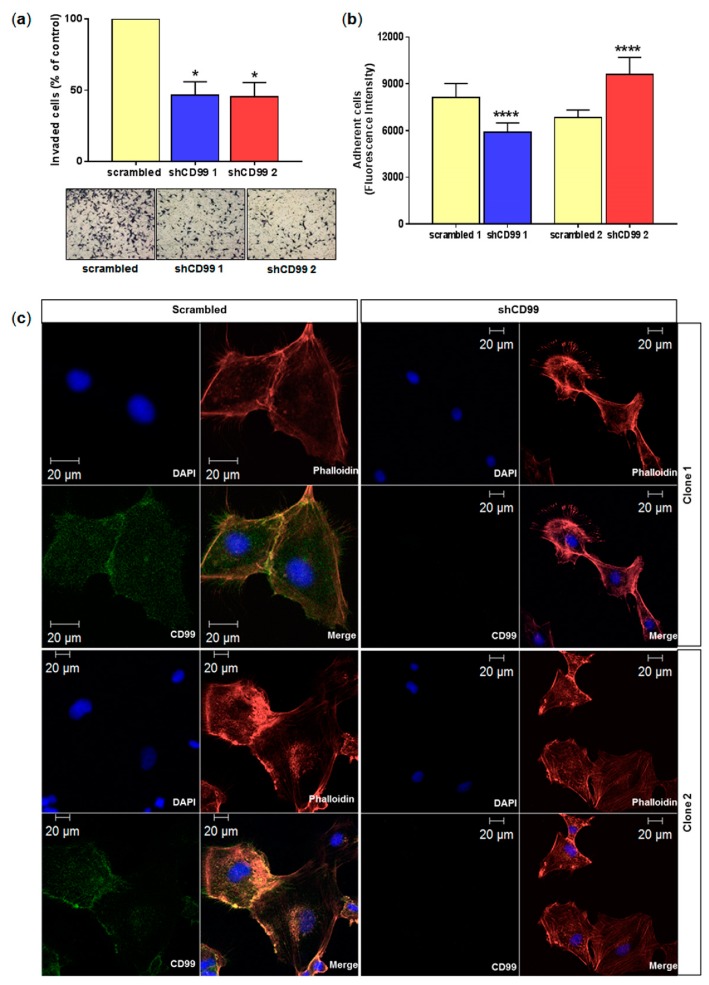
Role of CD99 in invasion and adhesion of the U87MG cell line, and colocalization of CD99 and phalloidin. (**a**) Trans-well invasion assay showing decreased U87MG invasion after CD99 knockdown (shCD99-1 and 2), when compared to the controls (scrambled). Graph represents percentage of invaded cells in relation to the control (means ± standard deviations of the means) in two independent experiments conducted in duplicate (* *p* < 0.05, Mann-Whitney test). The images show representative fields of U87MG cells that invaded and crossed the inserts (40x magnification). (**b**) Adhesion assay, with U87MG cells viability measurement by evaluation after 3 h of cell seeding onto the plate. The shCD99-1 U87MG cells attached less to the plate, while shCD99-2 U87MG cells attached more compared to its controls. The graph expresses the average of sextuplicate of two independent experiment (**** *p* < 0.0001, Mann-Whitney test). (**c**) Immunofluorescence showing that CD99 (green) and phalloidin (red) colocalize at cell-cell junctions and lamellipodia. Nuclei were stained with DAPI (blue). U87MG cell line knocked down for CD99 and controls. The images were analyzed by confocal microscopy by using a 40× objective.

**Figure 5 ijms-20-01137-f005:**
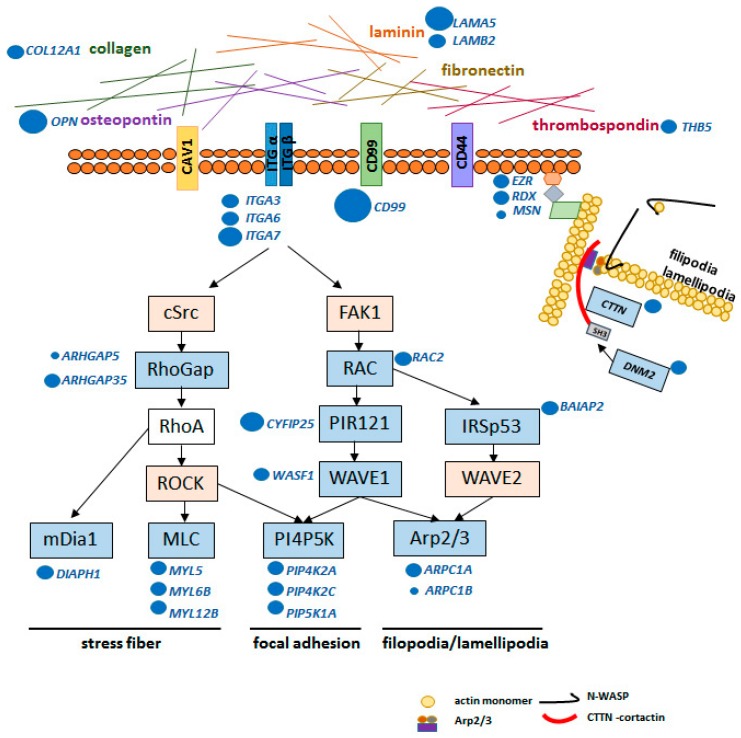
Schematic representation of the focal adhesion and actin regulation of cytoskeleton pathways, showing the plasma membrane and extracellular matrix target, and downstream pathways through c-Src and FAK1 with corresponding intermediate targets involved in stress fiber formation, focal adhesion, filopodia and lamellipodia formation. The gene symbol or the symbol for a group of genes are represented by a blue rectangle when downregulated or an orange rectangle when upregulated, according to the *CD99*-siRNA U87MG transcriptome analysis. The specific downregulated genes of each group are annotated next to the corresponding rectangle, and the size of blue juxtaposed circles is proportional to RNA-Seq fold change. Below the plasma membrane, on right side, the ERM complex (ezrin-radixin-moesin, coded by *EZR*, *RDX* and *MSC*, respectively) is represented linked to actin polymer, and at the actin branching site, Arp2/3 complex—cortactin (*CTTN*)—N-WASP—dynamin 2 (*DNM2*) are represented in lamellipodia and filopodia formation. All components were downregulated when *CD99* was silenced, demonstrating their role in the reduction of cell migration.

**Table 1 ijms-20-01137-t001:** PCR product size and sequence of primers for real time PCR.

Gene	PCR Product (bp)	Orientation	*Primer* (5′–3′)
*CD99* isoform 1	106	Forward	GATTGTGGGGGCTGTCGT
		Reverse	CACCTCCCCTTGTTCTGCATT
*CD99* isoform 2	108	Forward	GATTGTGGGGGCTGTCGT
		Reverse	TCCCTAGGTCTTCAGCCATCATT
*GUSB*	101	Forward	GAAAATACGTGGTTGGAGAGCTCATT
		Reverse	CCGAGTGAAGATCCCCTTTTTA
*HPRT*	118	Forward	TGAGGATTTGGAAAGGGTGT
		Reverse	GAGCACACAGAGGGCTACAA
*BCRP*	67	Forward	CCTTCGACGTCAATAACAAGGAT
		Reverse	CCTGCGATGGCGTTCAC

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
