# Peer review of "CD99 Expression in Glioblastoma Molecular Subtypes and Role in Migration and Invasion"

_ijms, 2019, doi:10.3390/ijms20051137_

Round 1
Reviewer 1 Report
In this study is addressed the role of CD99 in GBM invasiveness, showing that the expression of the isoform 1 controls is overexpress in gliomas as compared to non-tumor brain tissues. Moreover, it was identified, analyzing the TCGA database, a higher expression on the GBM with poorer prognosis (mesenchymal and classical vs. proneural). CD99 cellular evaluation was performed in U87 cells after gene silencing, demonstrating a reduced migratory and invasiveness behavior, supported by a transcriptome analysis showing a likely role of the integrin system.
The study is generally well performed and provide the readers with relevant information about a possible mechanism of GBM cell spreading. However, to allow publication the following points need to be addressed.
1) Please specify which CD99 isoform was evaluated in fog 1 b and c (I suppose it is 1 but this should be clarified). Moreover, in fig 1a, b and c median values, rather means should be indicated; in fig 1a statistical analysis should also be performed among the different glioma grades and not only vs. non tumor tissue.
2) The explanation provided for the unexpected result reported in fig 4B is not convincing, since it is not understandable why there are MORE adherent cells than in scr tx cells
MINOR POINTS
In fig 3c and e the label “invaded area” can be confusing with the invasion assay, please correct.
There are few misspelling (for ex. temozolamide, page 1; blade page 2), please correct.
Author Response
Dear Reviewer,
We appreciated the careful review of our manuscript and the suggestions to improve it. We took into consideration your comments and we made the modifications as suggested. We let in red all the alteration in the manuscript.
Thank you again for your attention on this matter.
Best regards,
Sueli Mieko Oba Shinjo

Reviewer 2 Report
The manuscript by Cardoso et al evaluated the differential expression of the levels of CD99 isoforms across human derived astrocytoma samples, and the CD99 related pathways in glioma U87 cells. The whole transcriptome analysis in U87 cells treated with siRNA CD99 showed a positive correlation between CD99 silencing and downregulation of genes related to cell motility regulation, being confirmed later with functional assays, when U87MG CD99-shRNA presented decreased migration and invasion rates.
Some analysis decisions and results need to be further clarified in order to properly assess the study validity.
1- A previous work by Seol et al showed that changes in CD99 expression increased U87 cell migration and invasiveness, and the authors explained that U87 cells were selected for CD99 overexpression experiments because it expressed almost no CD99. In contrary, in the present study, CD99 isoform was detected in U87 cell line. How can you explain such differences?
2- Regarding to U87 cell identified as glioblastoma (lines 18 and 246), please change to glioma model, once that it has been shown that the transcriptomes of “classical” glioma cell lines grown in serum containing medium (including U87MG) are different from those of glioma tissues, being thus poor representatives of the tumor of origin.
3- Please, fix temozolomide (it is misspelled in line 36).
4- In Figure 3b, looking at scrambled 1 and 2, looks that the right membrane is overexposed. Please, fix it to match both membranes exposure times.
5- What was the purpose of achieving a permanent long-term CD99 silencing using shRNA, instead of siRNA, if the last time-point observed was 24h? Was due to any concerns about cell viability after transfection using siRNA?
6- What are the main differences between Figures 1c and 1d? They look the same data, only arranged differently.
Author Response

(The authors gave the same response as above.)

Reviewer 3 Report
Comments:
Based on their previous work and through TCGA database analyses as well as staining of biopsy tissues from patients, the authors analyzed the differential expression of the two isoforms of CD99 in human astrocytoma specimens, and its involved signaling pathways in U87MG GBM cell line. The authors provided strong in vitro data in multiple models showing the important roles of CD99 in GBM molecular subtypes, whole transcriptome by RNA-Seq, and functional in vitro assays in CD99-shRNA in U87MG cells. The authors further demonstrated that astrocytoma of different malignant grades and U87MG cells only expressed CD99 isoform 1, which was higher in mesenchymal and classical than in proneural GBM subtypes. Genes related to actin dynamics, predominantly to focal adhesion, and lamellipodia/filopodia formation were downregulated in the transcriptome analysis if CD99 was silenced with siRNA. They also observed that decrease in tumor cell migration/invasion, and dysfunction of focal adhesion in functional assays. These findings offer new insight into the molecular parameters regarding GBM prognosis and response to therapy, and mechanisms utilized by CD99. If it can serve as a therapeutic target it may improve resectability and decrease the recurrence rate of GBM by decreasing tumor cells migration and invasion.
In general, this manuscript is technically sound and scientifically valid. The methods used are appropriate and properly conducted, and the conclusions drawn is convincing and supported by the experimental data presented. Sufficient methodological details were provided that the experiments could be reproduced.
Overall, this is an interesting study that is clearly presented, provides a strong rationale to pursue further study. Data are novel and may have a translational/therapeutic impact.
However, there are some weaknesses that can be readily addressed.
Considerations:
1. The mechanisms work heavily depended on a single cell line, U87MG. However, there are plenty of doubts have been expressed regarding U87MG's value in glioblastoma research (see below for reference). At least part of the work should be repeated in one of the cultures primarily from patient tissues.
http://www.the-scientist.com/?articles.view/articleNo/46929/title/Popular-Tumor-Cell-Line-Contaminated/
http://www.nature.com/news/venerable-brain-cancer-cell-line-faces-identity-crisis-1.20515
http://retractionwatch.com/2016/08/31/widely-used-brain-tumor-cell-line-may-not-be-what-researchers-thought-it-was/
2. The dual roles of CD99 were not clearly discussed. It was reported that engagement of CD99 induces the death of malignant cells through non-conventional mechanisms. In Ewing sarcoma, triggering of CD99 by specific monoclonal antibodies activates hyperstimulation of micropinocytosis and leads to cancer cells killing through a caspase-independent, non-apoptotic pathway resembling methuosis. This process is characterized by extreme accumulation of vacuoles in the cytoplasmic space.
Minor Comments
1. Figure 4.c is not very clear and does not show distinct details in the background. Because this can be very helpful to the reader, the author should replace this with a higher resolution photograph.
Author Response

(The authors gave the same response as above.)

Round 2
Reviewer 2 Report
Please, refer U87MG cells as glioma (not mesenchymal GBM) in discussion and methods (lines 299 and 311, respectively).
Author Response
Point 1: Please, refer U87MG cells as glioma (not mesenchymal GBM) in discussion and methods (lines 299 and 311, respectively).
Response 1: Both modifications were done.
Reviewer 3 Report
The authors have addressed satisfactorily the points I raised previously, in light of the response to both reviewers. With the additional data presented and discussion, the paper has substantially improved.
Author Response
Thank you very much for your comments, which helped us imporving the manuscript.